# Movement of Southern European Aquatic Alien Invertebrate Species to the North and South

Aldona Dobrzycka-Krahel 

Business Faculty, WSB Merito University in Gdańsk, Al. Grunwaldzka 238 A, 80-266 Gdańsk, Poland; adobrzycka@wsb.gda.pl

**Abstract:** Due to globalisation and anthropopressure (intensification of shipping, creation of water corridors connecting seas, cultivation of commercial species), the movement of aquatic species has increased in recent years. The determination of trends in the movement of aquatic species in their geographical distribution over time is important because it may help in the management of a species in aquatic ecosystems. There are also knowledge gaps on the long-term trends in the movements of Southern European aquatic alien invertebrates. The study provides the first evidence of both northward and southward movements of these species based on available observations from 1940 to 2021, using meta-analyses and GAM modelling. To date, the majority (98%) of analysed Southern European aquatic alien invertebrates of Mediterranean and Ponto-Caspian origin have moved to the north. Among them, 61% are Ponto-Caspian aquatic alien invertebrates that moved only to the north, and 4% are Mediterranean aquatic alien invertebrates that moved only to the north; the rest include species that moved to the north and south: 27% are Ponto-Caspian aquatic alien invertebrates, and 6% are Mediterranean aquatic alien invertebrates. The one-way movement to the south was observed only in 2% of Mediterranean aquatic alien species. The study will help in understanding the movement patterns of Southern European aquatic alien invertebrates and in the effective management of aquatic ecosystems that allow for the co-existence of people and the rest of biodiversity.

**Keywords:** global changes; movement of species; Mediterranean and Ponto-Caspian aquatic alien invertebrates; climate change

## 1. Introduction

There is often a knowledge gap about where and why species move. This knowledge is important to understand species distribution patterns [1,2]. Animals usually move to find more favourable conditions [3,4]. Many drivers affect the movement of organisms, e.g., climate change, disturbances in the natural habitats of organisms, etc. [5]. Among these drivers, shipping is considered the largest vector for the movement of aquatic species across the globe [6]. In the XVIIIth and XIXth centuries, important new waterways were opened. Numerous canals connecting the Mediterranean and the Ponto-Caspian areas with other parts of Europe were created as a result of industrial and economic human activity. Man-made interconnections of river basins (water corridors) have caused the movement of many aquatic species in Europe [7,8]. Thanks to these connections, the movement of, e.g., *Dreissena polymorpha* (Pallas, 1771) has started with climate suitability and the ability of species to successfully establish themselves [9], influencing the distribution of species [10].

It is common for species to move into cooler areas to escape warming [11,12]. Many studies have shown the northward movement of organisms, e.g., [13–15]. However, northward movement is usually considered in the context of native species, e.g., [16], despite the fact that human-mediated movement of species to the north has also been documented, e.g., [17–21]. Global warming is increasing, and it is estimated that by the end of the century, the average temperature on Earth will rise by 2.7 °C [22]. Temperature is a major factor in determining the geographical distribution of species [23–28]. Globalisation facilitates the

movement of species. But are alien species moving only northward in a changing world? Is there only one trend in movement?

The movement of species may have unpredictable consequences for ecosystems, and it is not possible to predict whether changes in the distribution will have a positive or negative effect. It is very important in species conservation and management to incorporate species movements into management objectives. This knowledge can be used to develop management strategies that may improve the effectiveness of management actions [2]. Animal movement is a core component of an ecosystem and may be vital for sustaining ecosystem processes such as trophic and species interactions [29–31]. Aquatic invertebrates constitute a significant diet source for many fish and water birds [32], so changes in their distribution may have important consequences for consumers. Some of these species are invasive (e.g., *Dikerogammarus villosus* (Sowinsky, 1894)) [33,34], so their possible presence and coexistence with other species [35], as well as a possible replacement of natives and/or threat to humans [36,37], may be important information for management actions.

However, there are knowledge gaps in documented evidence of long-term changes in the distribution of Southern European aquatic alien invertebrate species based on available observations, so the aim of this study is to analyse movement in the geographical distribution of these species.

The study addresses the gaps in knowledge by:

(1) analyses of long-term trends in the movement of Mediterranean and Ponto-Caspian aquatic alien invertebrate species;

(2) discussion on the responses of Southern European aquatic alien invertebrate species to changing conditions and management implications.

## 2. Materials and Methods

The list of all aquatic invertebrate species from the database GRIIS (Global Register of Introduced and Invasive Species) [38] with habitat and country of occurrence was downloaded. The geographical origin of aquatic invertebrate taxa was searched in the literature and in the databases AquaNIS (Information System On Aquatic Non-indigenous and Cryptogenic Species) [39] and GBIF (Global Biodiversity Information Facility) [40]. All Southern European aquatic alien invertebrate species (those living in freshwater, less than 0.5 ppt; brackish, 0.5–30 ppt; and marine waters, greater than 30 ppt) [41] of Mediterranean and Ponto-Caspian origin were selected.

Later, the occurrence records of Mediterranean and Ponto-Caspian aquatic alien invertebrate species were collected from GBIF [40]. Spatial records from GBIF are commonly used in decision-making processes and large-scale biogeography research [42].

Thus, the available occurrence data from 1940 to 2021, depending on research/ monitoring/reporting efforts, were processed. The trends of movement over time were determined using the maximum latitude in a year (northern extent) and/or the minimum latitude in a year (southern extent) (among all records of distributions in the GBIF database) of Mediterranean and Ponto-Caspian aquatic alien invertebrate species. Using occurrence data (degrees), a generalised additive model (GAM) approach was used to determine changes in the maximum/minimum latitude of occurrence of a species in the analysed years for evidence of species movement. The GAM models provide a useful tool to visualise the nature of the relationship between changes in the geographical distribution of a species over time. Latitudinal movements of the northern and southern limits were noticed (for species that moved north and south). For changes in geographical occurrences of particular species over time, the gamma distribution with log link function and categorisation was used. Estimates of movement (if sufficient data were available) were prepared due to the described methodology [43,44]. This GAM-based analysis was prepared using the Statistica 10 version. The shapes of the movements for the species were plotted. Additionally, the direction of species movement (northward and/or southward) was indicated based on information based on country of occurrence from GRIIS (Global Register of Introduced and Invasive Species) [38].

## 3. Results

*3.1. General Results on the Movement of Southern European Aquatic Alien Invertebrate Species*

The geographical occurrences of Southern European aquatic alien invertebrate species were analysed, and directions of movement were determined (Figure 1; Tables 1–4), depending on available data and research/monitoring/reporting efforts. Despite the fact that geographical distribution data are incomplete in GBIF, changes in the geographical distribution of Southern European aquatic alien invertebrate species over time were observed (Figures 2–4). To date, we have observed that the majority (98%) of the analysed Southern European aquatic alien invertebrate species (Mediterranean and Ponto-Caspian aquatic alien invertebrates) moved to the north. Among them, 61% are Ponto-Caspian aquatic invertebrates that moved only to the north and 4% are Mediterranean aquatic alien invertebrates that moved only to the north; the rest include 27% of Ponto-Caspian aquatic alien invertebrates that moved to the north and south and 6% of Mediterranean aquatic alien invertebrates that moved to the north and south. The one-way movement into the south (into warmer areas) was observed only in the case of 2% of species from Southern Europe (only Mediterranean aquatic alien invertebrates).

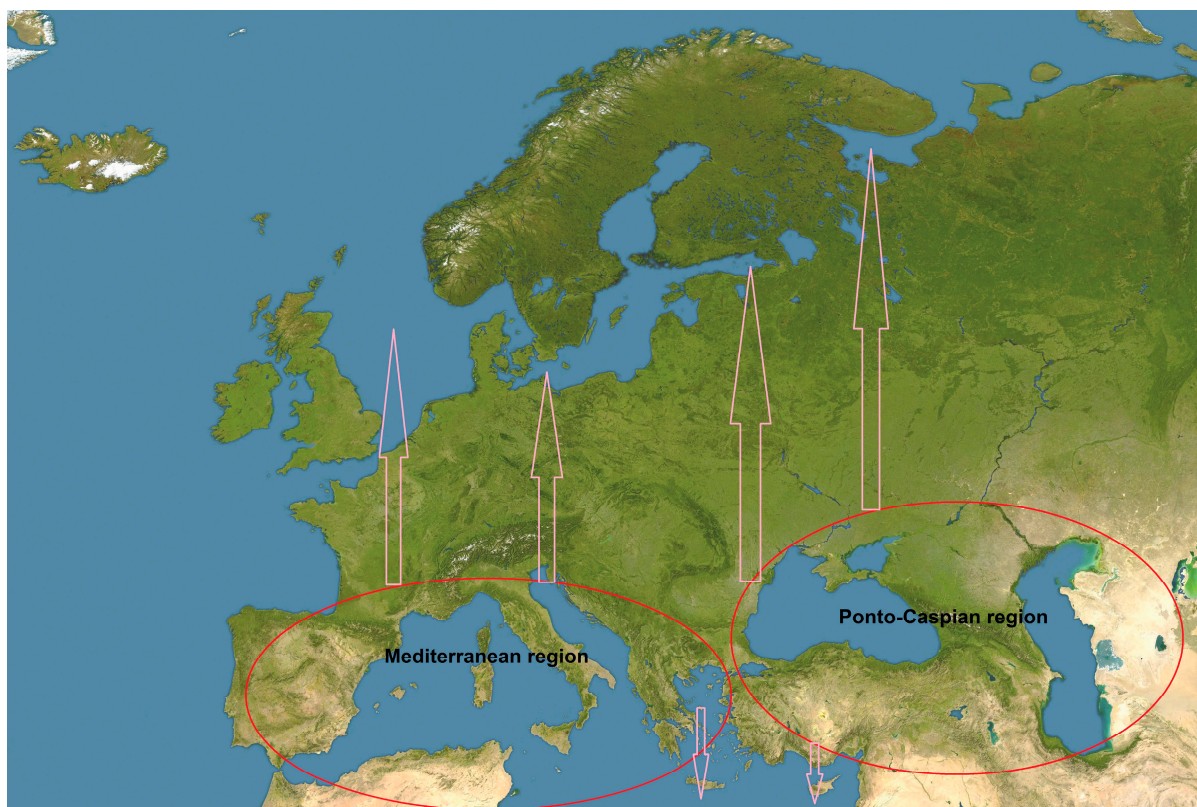

**Figure 1.** Location of the Mediterranean and Ponto-Caspian regions and directions of movements of species.

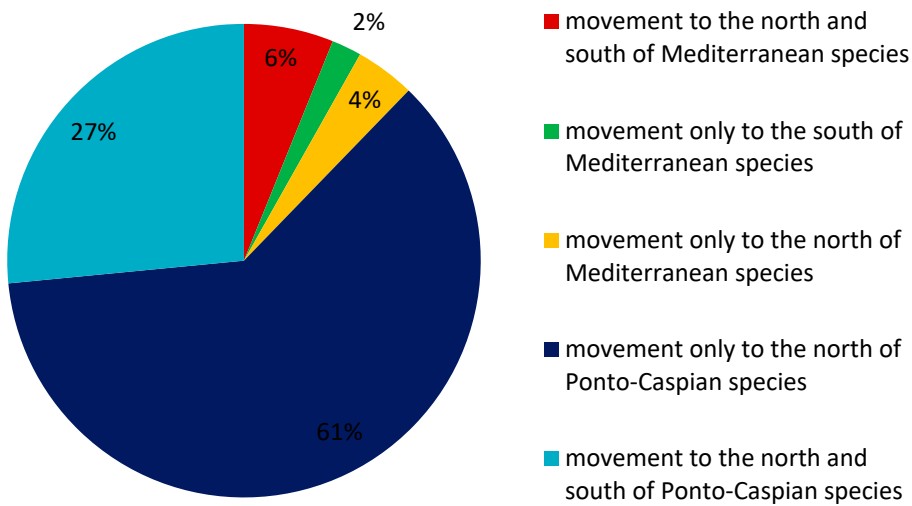

**Figure 2.** Movement of Southern European aquatic alien invertebrate species to the north and south (based on data from GBIF (Global Biodiversity Information Facility) [38] and GRIIS (Global Register of Introduced and Invasive Species) [40].

**Figure 3.** *Cont.*

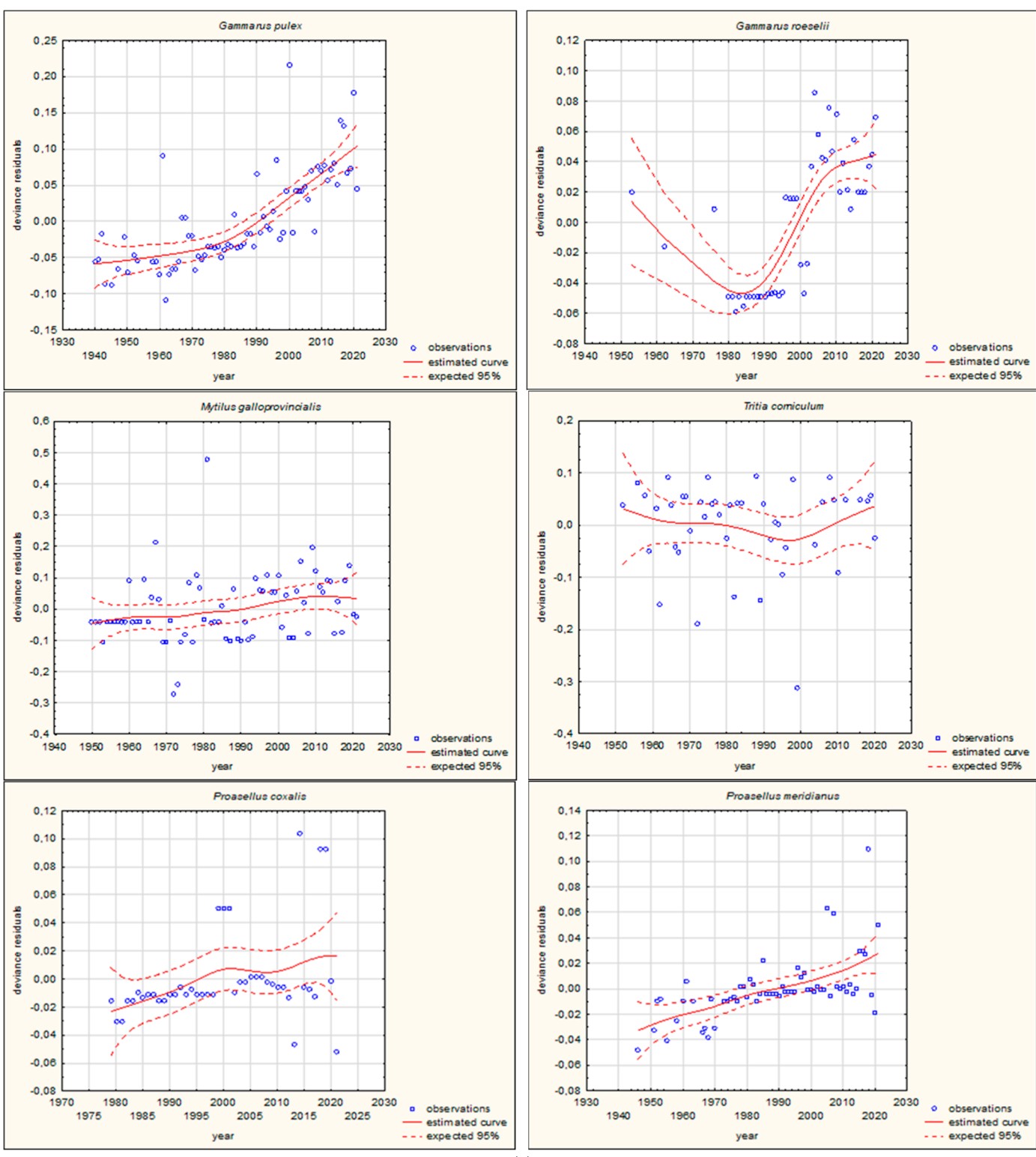

(**a**)

**Figure 3.** *Cont*.

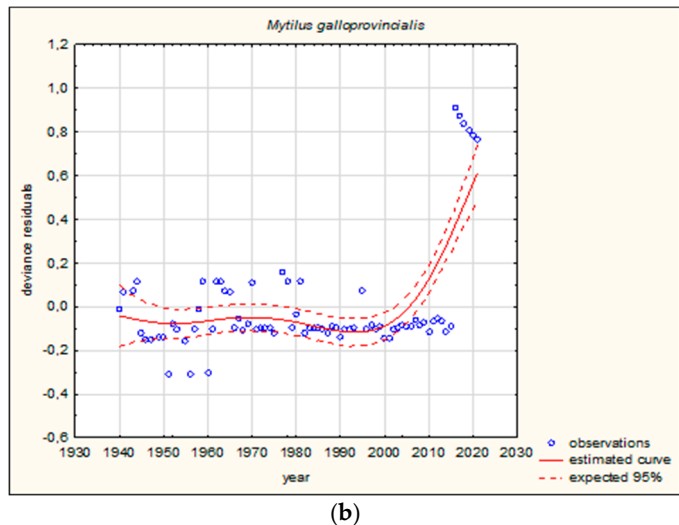

(**b**)

**Figure 3.** GAM models presenting curves of movement over time of Mediterranean aquatic alien invertebrates, based on GBIF (Global Biodiversity Information Facility) [38]. (**a**) Northward movement; (**b**) Southward movement.

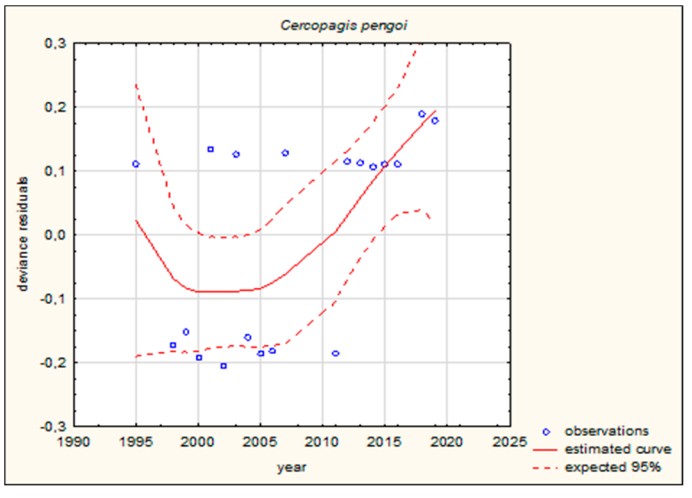

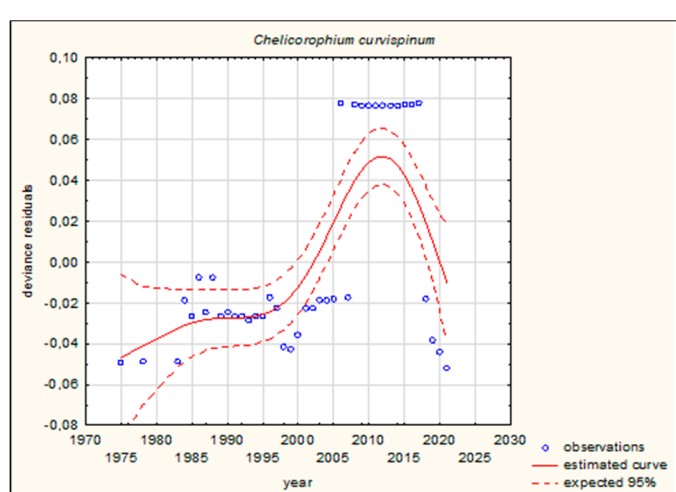

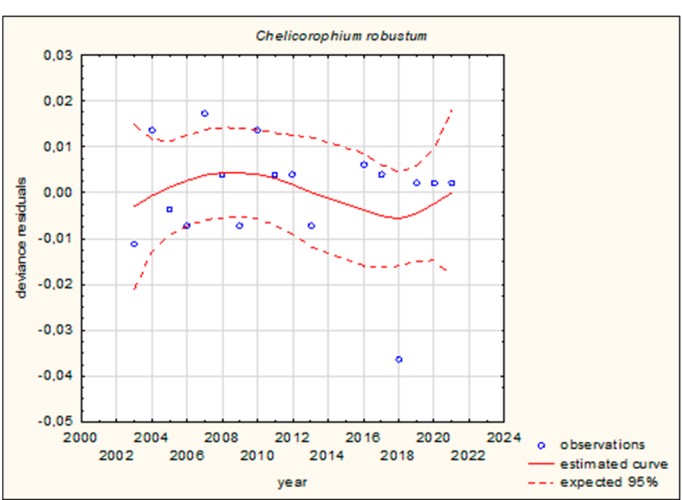

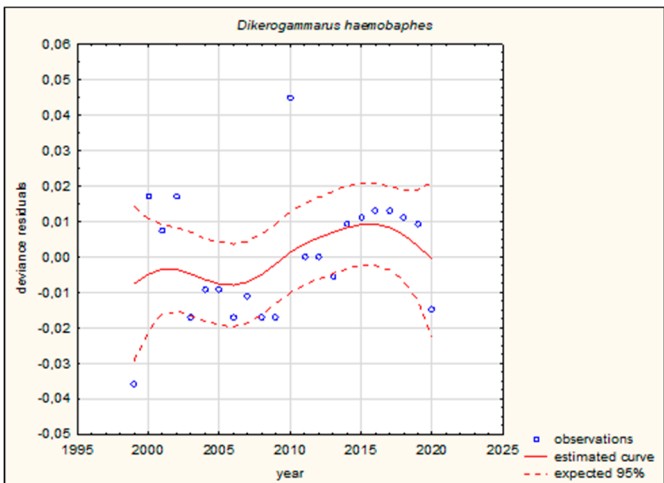

**Figure 4.** *Cont*.

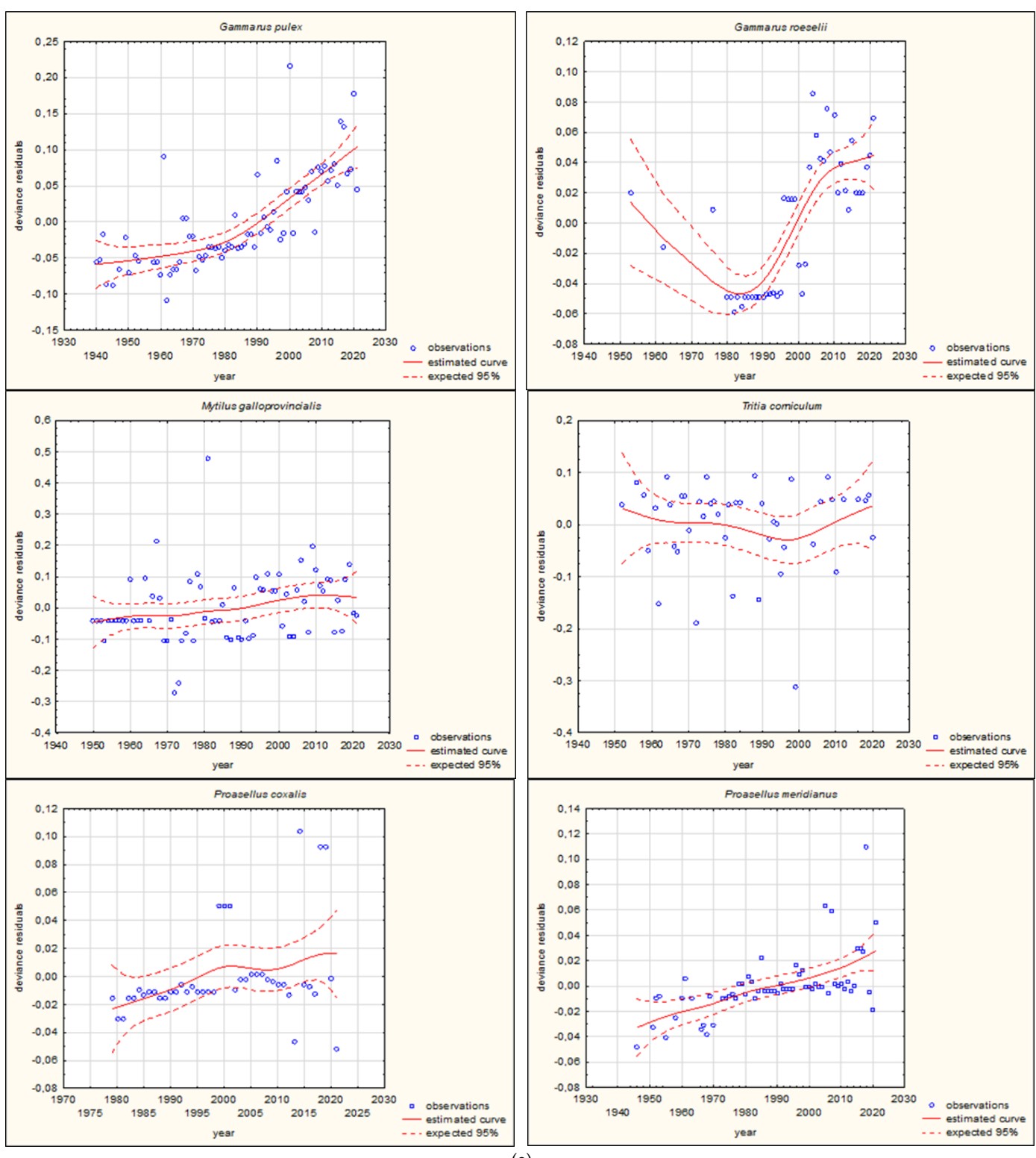

(**a**)

**Figure 4.** *Cont.*

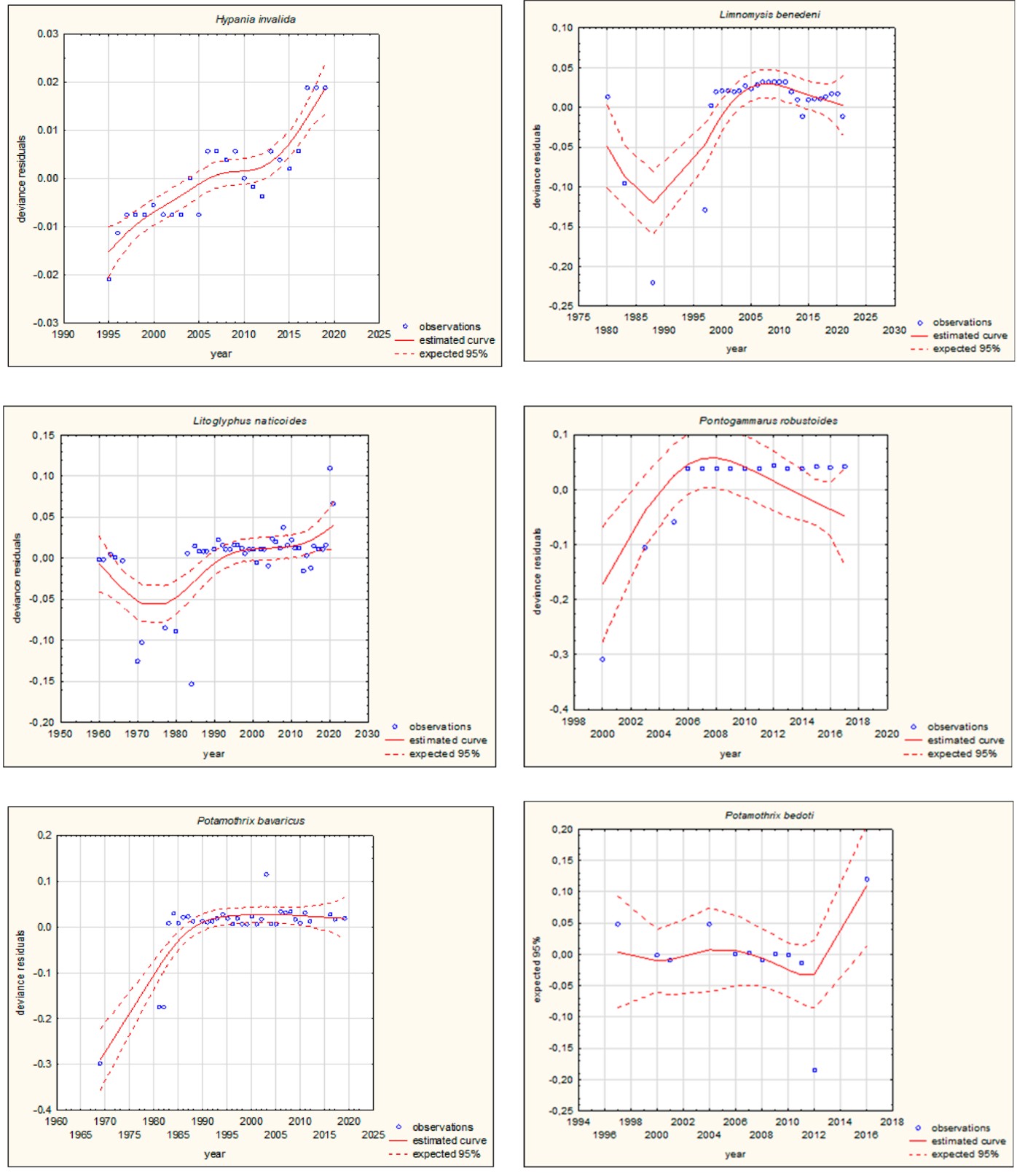

**Figure 4.** *Cont.*

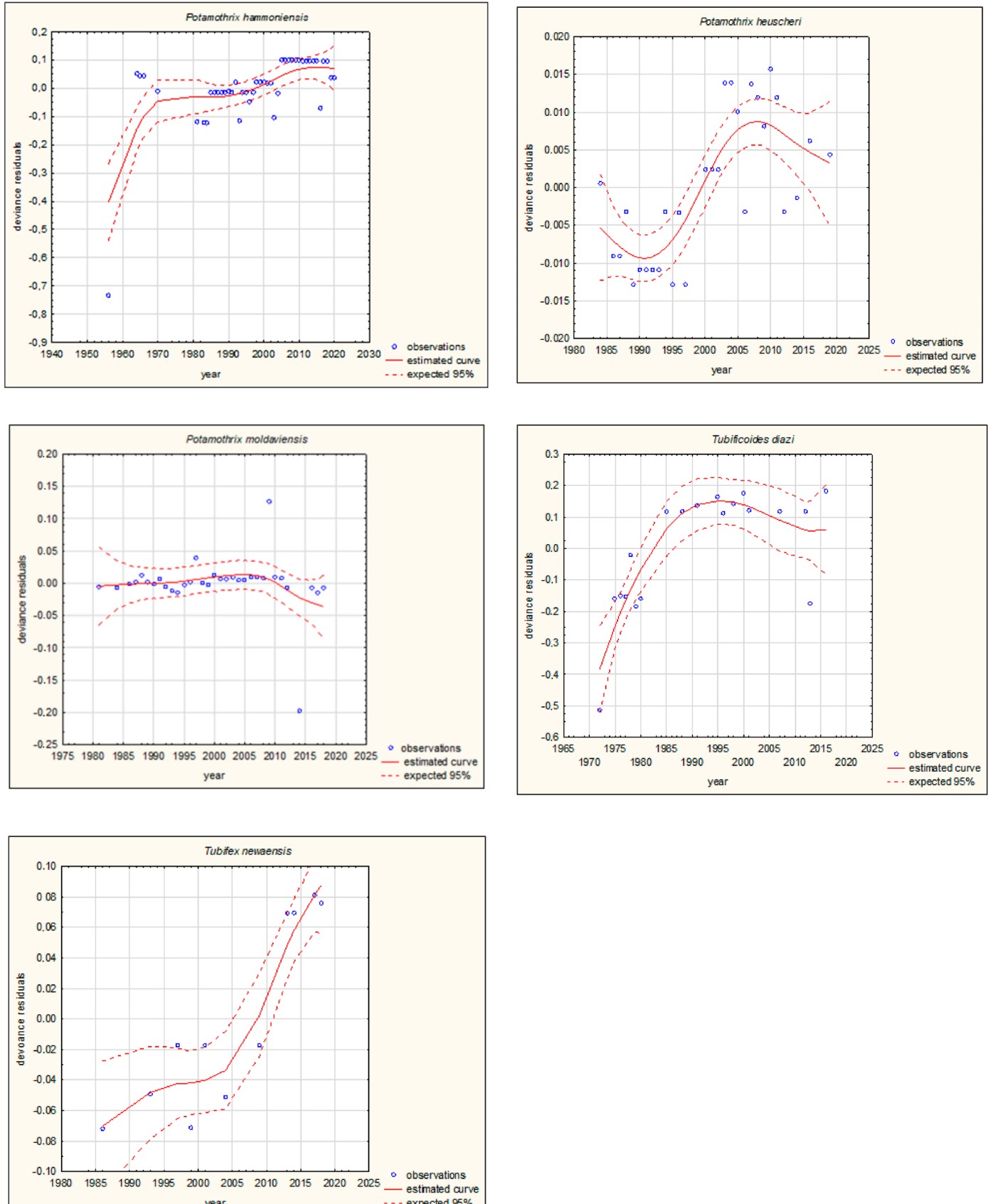

**Figure 4.** GAM models presenting curves of movement over time of Ponto-Caspian aquatic alien invertebrates, based on GBIF (Global Biodiversity Information Facility) [38]. Northward movement.

**Table 1.** Mediterranean aquatic alien invertebrate species that are characterised by movement, based on GBIF (Global Biodiversity Information Facility) [38] and GRIIS (Global Register of Introduced and Invasive Species) [40].

| | Species | Group | Movement ↑ Northward ↓ Southward | Pathway of Introduction |
|---|---|---|---|---|
| 1 | *Atyaephyra desmarestii* (Millet, 1831) | Crustacea | ↑↓ | Corridor |
| 2 | *Brachynotus sexdentatus* (Risso, 1827) | Crustacea | ↑↓ | NA |
| 3 | *Gammarus pulex* (Linnaeus, 1758) | Crustacea | ↑↓ | NA |
| 4 | *Gammarus roeselii* (Gervais, 1835) | Crustacea | ↑ | Corridor |
| 5 | *Echinogammarus berilloni* (Catta, 1878) | Crustacea | ↑ | Corridor |
| 6 | *Proasellus coxalis* (Dollfus, 1892) | Crustacea | ↑ | Corridor, stowaway, hull, fouling |
| 7 | *Proasellus meridianus* (Racovitza, 1919) | Crustacea | ↑ | Corridor, stowaway, hull, fouling |
| 8 | *Aporrhais pespelecani* (Linnaeus, 1758) | Mollusca | ↑↓ | NA |
| 9 | *Bela menkhorsti* (van Aartsen, 1988) | Mollusca | ↓ | NA |
| 10 | *Bogia labronica* (Bogi, 1984) | Mollusa | ↓ | NA |
| 11 | *Mytilus galloprovincialis* (Lamarck, 1819) | Mollusca | ↑↓ | Aquaculture |
| 12 | *Tritia corniculum* (Olivi, 1792) | Mollusca | ↑↓ | NA |

Abbreviation: NA—not available.

**Table 2.** Ponto-Caspian aquatic alien invertebrate species that are characterised by movement, based on GBIF (Global Biodiversity Information Facility) [38] and GRIIS (Global Register of Introduced and Invasive Species) [40].

| | Species | Group | Movement ↑ Northward ↓ Southward | Pathway of Introduction |
|---|---|---|---|---|
| 1 | *Amathillina cristata* (G.O.Sars, 1894) | Crustacea | ↑ | NA |
| 2 | *Amathillina pusilla* (G.O. Sars, 1896) | Crustacea | ↑ | NA |
| 3 | *Cardiophilus marisnigrae* (Miloslawskaya, 1931) | Crustacea | ↑ | NA |
| 4 | *Caspiocuma campylaspoides* (G.O. Sars, 1897) | Crustacea | ↑ | NA |
| 5 | *Cercopagis pengoi* (Ostroumov, 1891) | Crustacea | ↑ | Canals, shipping-fouling, ballast waters |
| 6 | *Chaetogammarus placidus* (G.O. Sars, 1896) | Crustacea | ↑ | NA |
| 7 | *Chaetogammarus warpachowskyi* (Sars, 1897) | Crustacea | ↑ | Deliberate with fish/shellfish |
| 8 | *Chelicorophium chelicorne* (G.O. Sars, 1895) | Crustacea | ↑ | NA |

**Table 2.** *Cont.*

|  | Species | Group | Movement ↑ Northward ↓ Southward | Pathway of Introduction |
|---|---|---|---|---|
| 9 | *Chelicorophium curvispinum* (G.O. Sars, 1895) | Crustacea | ↑ | NA |
| 10 | *Chelicorophium maeoticum* (Sowinsky, 1898) | Crustacea | ↑ | NA |
| 11 | *Chelicorophium nobile* (G.O. Sars, 1895) | Crustacea | ↑ | NA |
| 12 | *Chelicorophium mucronatum* (G.O. Sars, 1895) | Crustacea | ↑ | NA |
| 13 | *Chelicorophium robustum* (G.O. Sars, 1895) | Crustacea | ↑↓ | NA |
| 14 | *Chelicorophium sowinskyi* (Martynov, 1924) | Crustacea | ↑ | NA |
| 15 | *Compactogammarus compactus* (G.O. Sars, 1895) | Crustacea | ↑ | NA |
| 16 | *Cornigerius bicornis* (Zernov, 1901) | Crustacea | ↑ | NA |
| 17 | *Cornigerius lacustris* (Spandl, 1923) | Crustacea | ↑ | NA |
| 18 | *Cornigerius maeoticus* (Pengo, 1879) | Crustacea | ↑ | Canals |
| 19 | *Dikerogammarus bispinosus* (Martynov, 1925) | Crustacea | ↑ | Corridor, vessels |
| 20 | *Dikerogammarus haemobaphes* (Eichwald, 1841) | Crustacea | ↑↓ | Corridor, vessels |
| 21 | *Dikerogammarus villosus* (Sowinsky, 1894) | Crustacea | ↑↓ | Corridor, vessels |
| 22 | *Ectinosoma abrau* (Krichagin, 1877) | Crustacea | ↑ | NA |
| 23 | *Echinogammarus ischnus* syn. *Chaetogammarus ichnus* (Stebbing, 1899) | Crustacea | ↑ | Corridor, vessels |
| 24 | *Euxinia sarsi* (Sowinsky, 1898) | Crustacea | ↑ | NA |
| 25 | *Echinogammarus trichiatus* (Martynov, 1932) | Crustacea | ↑↓ | Corridor |
| 26 | *Echinogammarus warpachowskyi* (G.O. Sars, 1894) | Crustacea | ↑ | NA |
| 27 | *Euxinia weidemanni* (G.O. Sars, 1896) | Crustacea | ↑ | NA |
| 28 | *Evadne anonyx* (G.O. Sars, 1897) | Crustacea | ↑ | Canals, shipping |
| 29 | *Hemimysis anomala* (G.O. Sars, 1907) | Crustacea | ↑↓ | Stowaway, ballast water |
| 30 | *Heterocope caspia* (Sars G.O., 1897) | Crustacea | ↑ | NA |
| 31 | *Hypaniola kowalewskii* (Grimm and Annenkova, 1927) | Crustacea | ↑ | Fauna improvement |
| 32 | *Iphigenella acanthopoda* (G.O. Sars, 1896) | Crustacea | ↑ | NA |

**Table 2.** *Cont.*

|  | Species | Group | Movement ↑ Northward ↓ Southward | Pathway of Introduction |
|---|---|---|---|---|
| 33 | *Jaera istri* (Veuille, 1979) | Crustacea | ↑ | Corridor |
| 34 | *Jaera sarsi* (Valkanov, 1936) | Crustacea | ↑ | Canals |
| 35 | *Katamysis warpachowskyi* (G.O. Sars, 1893) | Crustacea | ↑ | Canals |
| 36 | *Kuzmelina kusnezowi* (Sowinsky, 1894) | Crustacea | ↑ | NA |
| 37 | *Lanceogammarus andrussowi* (G.O. Sars, 1896) | Crustacea | ↑ | NA |
| 38 | *Limnomysis benedeni* (Czerniavsky, 1882) | Crustacea | ↑↓ | Corridor |
| 39 | *Niphargoides corpulentus* (G.O. Sars, 1895) | Crustacea | ↑ | NA |
| 40 | *Niphargogammarus intermedius* (Carausu, 1943) | Crustacea | ↑ | NA |
| 41 | *Niphargus hrabei* (S. Karaman, 1932) | Crustacea | ↑ | NA |
| 42 | *Obesogammarus crassus* (G.O. Sars, 1894) | Crustacea | ↑↓ | Aquaculture |
| 43 | *Obesogammarus obesus* (G.O. Sars, 1894) | Crustacea | ↑ | Canals, vessels |
| 44 | *Paramysis lacustris* (Czerniavsky, 1882) | Crustacea | ↑↓ | Fisheries |
| 45 | *Paraniphargoides motasi* (Carausu, 1943) | Crustacea | ↑↓ | NA |
| 46 | *Pontogammarus robustoides* (Sars, 1894) | Crustacea | ↑ | NA |
| 47 | *Pontogammarus abbreviatus* (G.O. Sars, 1894) | Crustacea | ↑ | NA |
| 48 | *Pontogammarus aestuarius* (Derzhavin, 1924) | Crustacea | ↑ | NA |
| 49 | *Pontogammarus borceae* (Carausu, 1943) | Crustacea | ↑↓ | NA |
| 50 | *Pontogammarus maeoticus* (Sovinskij, 1894) | Crustacea | ↑↓ | NA |
| 51 | *Shablogammarus chablensis* (Carausu, 1943) | Crustacea | ↑ | NA |
| 52 | *Shablogammarus subnudus* (G.O. Sars, 1896) | Crustacea | ↑ | NA |
| 53 | *Stenogammarus carausui* (Derzhavin and Pjatakova, 1962) | Crustacea | ↑ | NA |
| 54 | *Stenogammarus compressus* (G.O. Sars, 1894) | Crustacea | ↑↓ | NA |
| 55 | *Stenogammarus macrurus* (Sars, 1894) | Crustacea | ↑ | NA |
| 56 | *Stenogammarus similis* (Sars, 1894) | Crustacea | ↑↓ | NA |
| 57 | *Turcogammarus aralensis* (Uljanin, 1875) | Crustacea | ↑ | NA |
| 58 | *Uroniphargoides spinicaudatus* (Carausu, 1943) | Crustacea | ↑ | NA |

**Table 2.** *Cont.*

| | Species | Group | Movement<br>↑ Northward<br>↓ Southward | Pathway of Introduction |
|---|---|---|---|---|
| 59 | *Yogmelina limana* (Karaman and Barnard, 1979) | Crustacea | ↑ | NA |
| 60 | *Abra segmentum* (Récluz, 1843) | Mollusca | ↑↓ | NA |
| 61 | *Dreissena polymorpha* (Pallas, 1771) | Mollusca | ↑↓ | NA |
| 62 | *Dreissena rostriformis bugensis* (Andrusov, 1897) | Mollusca | ↑↓ | Corridor, stowaway, contaminant |
| 63 | *Dreissena rostriformis* (Deshayes, 1838) | Mollusca | ↑ | NA |
| 64 | *Euxinipyrgula lincta* (Milaschewitsch, 1908) | Mollusca | ↑ | NA |
| 65 | *Hypanis colorata* (Eichwald, 1829) | Mollusca | ↑ | NA |
| 66 | *Hypanis pontica* (Eichwald, 1838) | Mollusca | ↑ | NA |
| 67 | *Hypanis fragilis* (Milaschevitch, 1908) | Mollusca | ↑ | NA |
| 68 | *Hypanis glabra* (Ostroumoff, 1905) | Mollusca | ↑ | NA |
| 69 | *Lithoglyphus naticoides* (C.Pfeiffer, 1828) | Mollusca | ↑↓ | Corridor |
| 70 | *Viviparus acerosus* (Bourguignat, 1862) | Mollusca | ↑ | Release, escape |
| 71 | *Blackfordia virginica* (Mayer, 1910) | Cnidaria | ↑↓ | NA |
| 72 | *Cordylophora caspia* (Pallas, 1771) | Cnidaria | ↑↓ | Stowaway, ballast water, hull, fouling |
| 73 | *Caspiobdella fadejewi* (Epshtein, 1961) | Annelida | ↑ | NA |
| 74 | *Hypania invalida* (Grube, 1860) | Annelida | ↑ | Stowaway, hull, fouling |
| 75 | *Hypaniola kowalewskii* (Grimm and Annenkova, 1927) | Annelida | ↑ | NA |
| 76 | *Isochaetides michaelseni* (Lastockin, 1937) | Annelida | ↑ | NA |
| 77 | *Potamothrix heuscheri* (Bretscher, 1900) | Annelida | ↑↓ | Corridor |
| 78 | *Potamothrix vejdovskyi* (Hrabe, 1941) | Annelida | ↑↓ | Corridor, stowaway, ballast water |
| 79 | *Potamothrix moldaviensis* Vejdovský and Mrázek, 1903 | Annelida | ↑ | NA |
| 80 | *Potamothrix bavaricus* (Oschmann, 1913) | Annelida | ↑↓ | NA |
| 81 | *Potamothrix bedoti* (Piguet, 1913) | Annelida | ↑↓ | NA |
| 82 | *Potamothrix hammoniensis* (Michaelsen, 1901) | Annelida | ↑↓ | NA |
| 83 | *Psammoryctides moravicus* (Hrabe, 1934) | Annelida | ↑↓ | Corridor, stowaway, ballast water |

**Table 2.** *Cont.*

| | Species | Group | Movement ↑ Northward ↓ Southward | Pathway of Introduction |
|---|---|---|---|---|
| 84 | *Tubificoides diazi* (Brinkhurst and Baker, 1979) | Annelida | ↑↓ | NA |
| 85 | *Tubifex newaensis* (Michaelsen, 1903) | Annelida | ↑ | NA |

Abbreviation: NA—not available.

**Table 3.** Statistics of GAMs of Mediterranean aquatic alien invertebrate species movement, including significance levels (*p*-values) and degrees of freedom (d.f.) only for species with significant differences at the 0.05 level.

| Species | Taxonomic Position | Statistics of Movement | | | |
|---|---|---|---|---|---|
| | | ↑ Northward *p*-Value | d.f. | ↓ Southward *p*-Value | d.f. |
| *Gammarus pulex* (Linnaeus, 1758) | Crustacea | 0.010010 | 4 | | |
| *Gammarus roeselii* (Gervais, 1835) | Crustacea | 0.0000001 | 4 | | |
| *Echinogammarus berilloni* (Catta, 1878) | Crustacea | 0.002880 | 4 | | |
| *Aporrhais pespelecani* (Linnaeus, 1758) | Mollusca | 0.014661 | 4 | | |
| *Mytilus galloprovincialis* (Lamarck, 1819) | Mollusca | 0.789822 | 4 | 0.0000001 | 4 |

**Table 4.** Statistics of GAMs of Ponto-Caspian aquatic alien invertebrate species movement, including significance levels (*p*-value) and degrees of freedom (d.f.) only for species with significant differences at the 0.05 level.

| Species | Taxonomic Position | Statistics of Movement | | | |
|---|---|---|---|---|---|
| | | ↑ Northward *p*-Value | d.f. | ↓ Southward *p*-Value | d.f. |
| *Chelicorophium curvispinum* (G.O. Sars, 1895) | Crustacea | 0.0000001 | 4 | | |
| *Limnomysis benedeni* Czerniavsky, 1882 | Crustacea | 0.000002 | 4 | | |
| *Dreissena polymorpha* (Pallas, 1771) | Mollusca | 0.048831 | 4 | | |
| *Dreissena rostriformis bugensis* (Andrusov 1897) | Mollusca | 0.005452 | 4 | | |
| *Lithoglyphus naticoides* (C. Pfeiffer, 1828) | Mollusca | 0.000627 | 4 | | |
| *Hypania invalida* (Grube, 1860) | Annelida | 0.014347 | 4 | | |
| *Potamothrix bavaricus* (Oschmann, 1913) | Annelida | 0.0000001 | 4 | | |
| *Potamothrix hammoniensis* (Michaelsen, 1901) | Annelida | 0.000046 | 4 | | |
| *Tubificoides diazi* (Brinkhurst and Baker, 1979) | Annelida | 0.000078 | 4 | | |

*3.2. Long-Term Trends in the Movement of Southern European Aquatic Alien Invertebrate Species*

The long-term tendency in the movement of Southern European aquatic alien invertebrate species to higher northern latitudes (observed in the majority—in 98% of species) shows that these species generally moved to cooler areas. But in the case of some species, movement to the south occurred. However, the number of geographical coordinates confirming the movement of species to the south was not sufficient for the preparation of the plot in GAM.

The shapes of the trends of movements over time in GAMs for each species (if sufficient data were available) are illustrated in Figures 3 and 4. Based on the changes in the maximum/minimum latitude over time, the movement of the analysed species to the north, towards higher latitudes, and/or to the south, towards lower latitudes, was confirmed. Interestingly, in the case of some Southern European alien aquatic invertebrates, only one-way movement to the south was recorded, but only in the case of the Mediterranean species *Mytilus galloprovincialis* was GAM analysis possible. When the deviance residuals were plotted against time (years), clear patterns were verified (Figures 3 and 4). The deviance residuals in the presented models show how well the movement of species is confirmed by the models and present a discrepancy between the observations and the estimated curves. The GAMs confirmed that movement of Southern European aquatic alien species occurs over time (Figures 3 and 4) and indicates that species display non-linear changes in distribution over time. However, a GAM analysis of a multiyear dataset might reveal that the movement in many cases is very low but in others is relatively high. Comparing the shapes of the plots (Figures 3 and 4), the movement patterns of species differ among various species depending on their origin. In the case of Mediterranean aquatic alien species, the shapes are more linear, but in the case of Ponto-Caspian aquatic alien species, they are usually more irregular.

## 4. Discussion

*4.1. Movement of Southern European Aquatic Alien Invertebrates*

The study provides the first evidence of long-term movements in the distribution of Southern European aquatic alien invertebrate species using meta-analysis and GAM-based modelling. The study demonstrates that GAM-based modelling could be used to create non-linear, spatial changes in the distribution of species over time. Moreover, GAM-based analyses better illustrate movement patterns than linear models, often used for confirmation of distribution trends [45]. The movement of many Southern European aquatic alien species was confirmed based on available data; for all the rest, there is a knowledge gap. The obtained results were based on the analysed period (from the 1940s to the 2000s) and depended on differences in research/monitoring/reporting efforts. Individual species' movements vary in their rates of change. Perhaps different shapes of movement could identify different responses of species towards environmental conditions, e.g., temperature, salinity, habitat degradation, etc., as a result of different introduction pathways.

The rate of settling of aquatic alien invertebrate species is influenced by a combination of morphological, behavioural, and ecological features [17,46]. A comprehensive understanding of these traits and interactions is crucial for predicting and managing the spread of aquatic alien species. Morphological adaptations can play a crucial role in the settlement success of alien aquatic invertebrates. Features such as body shape, size, and dispersal mechanisms can affect their ability to colonise new environments. For example, species with efficient dispersal structures such as specialised appendages or buoyancy adaptations may have a higher settlement rate compared to those lacking such traits [17,47]. Behavioural traits of aquatic alien invertebrates can significantly influence their ability to settle in new habitats. For instance, the ability to respond to a wide range of environmental conditions, different resource availability, and competitive interactions with native species can impact the success of settlement [19,36]. Behavioural traits related to, e.g., feeding habit, capacity for behavioural thermoregulation, and aggression can influence the establishment and spread of alien species [17,33,48]. Physiological traits, including quiescence and dor-

mancy (or diapause), which help to overcome adverse conditions, as well as tolerance of a wide range of temperatures, euryeocity, and a high reproduction rate, are important in the settlement success of alien species [17]. Understanding the ecological context in which aquatic alien invertebrates settle is also essential. The availability of suitable resources, including food, shelter, and reproductive sites, can greatly influence their establishment. Furthermore, interactions with native species, both competitive and facilitative, can play a pivotal role in determining the settlement rate of alien species [17,49]. Several case studies provide valuable insights into the relationship between morphological, behavioural, physiological, and ecological features and the settlement rate of alien aquatic invertebrate species. By examining specific examples, such as, e.g., the zebra mussels (*D. polymorpha*) and the killer shrimp (*D. villosus*), we can observe how their unique characteristics contribute to the rapid colonisation of new habitats [19,50,51].

By identifying the key morphological, behavioural, physiological, and ecological traits associated with the successful settlement of aquatic alien species, we can enhance early detection and implement targeted control measures to prevent or mitigate their negative impacts on native ecosystems [52].

Moreover, we should understand that aquatic alien invertebrates from Southern Europe gained the opportunity for movement through different pathways. As a consequence of shipping and the creation of navigable canals and waterways enabling connections of the Mediterranean and the Ponto-Caspian regions with other parts of Europe, their spread to the north was possible [7,8], as was movement to the south [53]. The northward movement was confirmed in this study by the majority of Southern non-indigenous aquatic species. This is in line with previous observations: Ponto-Caspianization of central and western European waterways [54]. Interestingly, some Southern European aquatic invertebrates moved to the south, into lower latitudes, where they most probably evolved elevated upper thermal limits relative to the species in northern latitudes, facilitating their establishment in warm water bodies [55] (Figure 2; Tables 1–4). Generally, it is well known that temperature variability imposes intensified peak stress [56]. However, detailed knowledge of which individuals and species are most likely to survive and why under upper thermal limits is poor, indicating that smaller individuals survived to higher temperatures than large animals and active species survived to higher temperatures than sessile or low-active species when temperatures were raised acutely [57]. Ecological generalists, with higher heat tolerances, are competitive at more extreme and increasing temperatures [58].

A major problem is that changes in the extent and impacts of invasions are occurring with the accumulation of impacts and through synergisms with other components of global change [59].

### 4.2. Responses of Southern European Aquatic Alien Invertebrates to Changing Conditions

The analysed species were able to move outside their original ranges of distribution. Most likely, increases in regional sea temperatures have triggered a major northward movement of species. Projected climate change in the Mediterranean and Ponto-Caspian areas with higher temperatures and increased periodic drought can be expected to further increase the instability of habitats [60–62]. Climate warming will accelerate the movement process and favour species movement from southern to northern latitudes in Europe [62] in search of more favourable thermal conditions compared with those existing in the original areas.

Southern aquatic species may have only a northern direction of movement if climate change is the only reason affecting movement. However, the latest studies report that another reason for species' movements is most likely the destruction of habitats they previously inhabited. In fact, biodiversity loss in the Mediterranean and Ponto-Caspian regions may be caused by habitat degradation, coastal infrastructural development, and damming of rivers, so biodiversity here is under threat [62,63]. Habitat modifications disturbed previous natural salinity gradients and settings in the Ponto-Caspian area [61]. In many places in the Ponto-Caspian area, native species have been replaced by invasive

species, e.g., *Mytilopsis leucophaeata* (Conrad, 1831), *Potamopyrgus antipodarum* (Gray, 1843), *Rhithropanopeus harrisii* (Gould, 1841), and other euryhaline species [62]. Similarly, the Mediterranean region is regarded as a hot spot for invasive alien species that tend to decline native species [64].

*4.3. Management Implications*

The results of this study may suggest several recommendations for management actions in areas of the introduction of Southern European aquatic invertebrate species. The Convention on Biological Diversity (CBD) prioritises preventing the introduction of alien invasive species and thereby avoiding adverse impacts. Firstly, on-going monitoring activities are recommended to record new species. Many Mediterranean and Ponto-Caspian aquatic invertebrates are easily adaptable to novel environments, e.g., [17,65]. Unfortunately, these species are generalists [65–68]—they utilise different food sources and have relatively wide thermal tolerance [69,70]. Some of these are tending to decline, compete, and displace native species from their habitats [71–73] and to re-engineer the new ecosystems [74]. Further changes in the distribution of Southern European aquatic alien invertebrate species are expected due to climate change and the degradation of habitats in their native areas. Once these species become established, management is difficult and economically costly [75]. It aims for species eradication, complete reproductive removal, containment, and/or population suppression [76]. But new invaders are nearly impossible to fully eradicate [77], so early identification of aquatic alien invaders in new areas should be prioritised because species management is not working well in the marine realm.

Pathogens and parasites are the next aspects connected with species movement. Associated life, e.g., bacteria, viruses, fungi, and other organisms, as well as different epibionts and endobionts, may be consequences of alien species introduction because alien species may carry new organisms [78,79], with unpredictable consequences for humans and other biota.

Therefore, understanding the movement of Southern European aquatic alien invertebrate species enables managers to identify threats and prioritise management actions. An important way to reduce introductions is to manage vectors and pathways [80]. As shipping is the primary pathway for the introduction of aquatic organisms, mainly invertebrates [81,82], it needs such management actions as, e.g., hull cleaning, antifouling, and ballast water exchange [83]. Such actions have the potential to reduce the establishment and spread of aquatic alien species. But not always mentioned interventions are sufficient. Dispersal corridors (water corridors among marine basins) are also considered introduction pathways and require responsibility [84–87]. Another pathway of introduction of aquatic alien species is cultivation, and some invertebrates, e.g., the commercial species of the mussel *Mytilus galloprovincialis* (Lamarck, 1819), are introduced in this way. The potential impact of the species once it is established in the new aquatic ecosystem is uncertain because it can outcompete native mussels and because of its high reproductive potential and adaptability to different environments [87]. Prevention measures for such species should include prohibiting cultivation in foreign areas and monitoring unintentional introductions.

Effective biological management demands complex multisectoral and multinational collaboration, and much work remains to be carried out. Success in such ventures holds the key to reducing the influx of alien species.

On the other hand, climate change and habitat degradation facilitate alien species movement [56,88] and should be limited.

**5. Conclusions**

Prevalent long-term trends in the movement of Southern European aquatic alien invertebrates to the north were confirmed. The movement of Southern European aquatic alien invertebrates to the south was also observed, but not on a sufficiently large scale. Maybe in the future, if the observed trend is continued, Southern European alien species will be common not only in the Northern Hemisphere [21], but also in the Southern one. Due to the potentially adverse impact of many aquatic invasive alien species moving to the north

and south, understanding changes in the geographical distribution of species has relevance to management efforts. Understanding the morphological, behavioural, physiological and ecological features associated with the settlement rate of aquatic alien species is vital for developing effective management strategies. Strong human impacts, including shipping, intensification of use of waterways, cultivation, climate change, and habitat degradation, should be limited in the future. For sustainable use of aquatic ecosystems, preventing actions (e.g., hull cleaning, antifouling, ballast water exchange) should be prioritised.

**Funding:** This research received no external funding.

**Institutional Review Board Statement:** Not applicable.

**Informed Consent Statement:** Not applicable.

**Data Availability Statement:** Not applicable.

**Conflicts of Interest:** The author declares no conflict of interest.

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
