# Peer review of "Movement of Southern European Aquatic Alien Invertebrate Species to the North and South"

_water, doi:10.3390/w15142598_

Round 1
Reviewer 1 Report
Based on data collected from online database and literature, this paper analyzes the trend in movements of South European alien aquatic invertebrates. The results are interesting and indicate that the main trigger of the northword movement might be the regional sea temperatures. However, there are several questions that might need to be addressed by the author: 1) there are quite a few grammatical errors occurred in the paper, and some sentences could be revised to make the author’s idea clearer, like line 11, 30, 49, 66, 92, 126, etc. 2) it is better that more detailed interpretation of figure 3&4 can be provided, like the meaning of y-axis ‘deviance residuals’. 3) since the author mentioned that there are several factors (temperature, salinity, habit degradation etc.) that can affect the movements of species, can the author collect data of these environmental factors at the same time when collect biotic data, and explore the exact reason that result in the northward movement?
There are quite a few grammatical errors occurred in the paper, and some sentences could be revised to make the author’s idea clearer.
Author Response
Dear Reviewer
thank you very much for your effort in evaluation of my manuscript
Best regards
Aldona Dobrzycka-Krahel

Reviewer 2 Report
This work is quite interesting. The text is well-written, in my opinion. However, small orthographical and stylistic errors still presented. Therefore, authors should carefully check the manuscript and make corrections. The «Introduction» is informative, giving a good insight about alien aquatic invertebrate species.
I recommend a number of corrections to the manuscript:
1. Although in «Materials and Methods» section is written that: "The geographical origin of aquatic invertebrate taxa were searched in the literature, in the database AquaNIS" and "All Southern European non-indigenous aquatic invertebrate species of Mediterranean and Ponto-Caspian origin were selected", but (!) as a specialist in Copepoda, I know for sure that Heterocope appendiculata, marked in Table 4 as Ponto-Caspian alien aquatic invertebrate species, has widely distribution in northern and even Arctic regions of Eurasia. I doubt this species is from Ponto-Caspian origin. The range of this species lies mainly in the northern part of Eurasia. It is possible that Heterocope appendiculata migrated to the northern regions of Europe, not from the south, but opposite from the northeast. I think that assigning of this species to Ponto-Caspian alien aquatic invertebrate is a mistake of database AquaNIS, that is unfortunately not uncommon for large databases. In this regard, I strongly recommend to the authors check the ranges of all other alien aquatic invertebrate species included in the analysis. The species Heterocope appendiculata should be excluded from the analysis.
2. In «Results» section is written: «Individual species movement vary in their rates of change.» It is highly desirable to include in the «Discussion» section a paragraph about what morphological, behavioral and ecological features are associated with rate of settling of alien aquatic invertebrate species. Which taxonomic groups of invasive species are dispersing faster and which are slower?
3. In Tables 3 and 4, only species with “significant differences at the 0.05 level in red” should be left. The article is already over saturated with tables and this is somewhat complicates for perception by the readers.
4. Presented article partly overlaps in some points with: Soto, I., Cuthbert, R.N., Ricciardi, A. et al. The faunal Ponto-Caspianization of central and western European waterways. Biol Invasions 25, 2613–2629 (2023). https://doi.org/10.1007/s10530-023-03060-0. However, this work is not cited by the authors, and this should be corrected. The article Soto et al., 2023 should be included in the analysis in the «Discussion» section and may be cited in the "Introduction" section.
In addition, in the "Discussion" in the analysis of the dispersal factors for invasive species is desirable cite: Kotov A.A.; Karabanov, D.P.; Van Damme, K. Non-Indigenous Cladocera (Crustacea: Branchiopoda): From a Few Notorious Cases to a Potential Global Faunal Mixing in Aquatic Ecosystems. Water 2022, 14, 2806. https://doi.org/10.3390/w14182806. Because Cladocera are among the most active and dangerous to water ecosystem alien species.
I recommend this manuscript for publication in “Water” after minor revision.
Author Response

(The authors gave the same response as above.)

Reviewer 3 Report
In the paper is highlighted as movement of aquatic species has increased in recent years and this study can help in understanding movement patterns of South European alien aquatic invertebrates. Effective management of aquatic ecosystems that allows for coexistence of people and the rest of biodiversity are taken into consideration.
This paper is relevant and presents methodological accuracies and appropriateness of references.
Figures and tables are understandable and well structured.
Minor editing of English language is required.
Author Response

(The authors gave the same response as above.)

Round 2
Reviewer 1 Report
There are still several grammatical problems in the paper, like line 11, 65, 192, 202, etc. There might be more problems existed, and thus the author should check the language of the whole paper.
In Table 3, it said "only for species with p-value less than 0.05", but p-value of the last species is 0.789822. Also, the caption of Table 3 & 4 should be revised.
Need to be checked and revised.
Author Response
Dear Reviewer
thank you very much for your effort, suggestions and advises
Best wishes
Aldona Dobrzycka-Krahel
